# Assessment of Physicochemical and Nutritional Characteristics of Waste Mushroom Substrate Biochar under Various Pyrolysis Temperatures and Times

**Rubab Sarfraz** , **Siwei Li, Wenhao Yang, Biqing Zhou and Shihe Xing *

Fujian Provincial Key Laboratory of Soil Environmental Health and Regulation, College of Resources and Environment, Fujian Agriculture and Forestry University, Fuzhou 350002, China; 2161907001@fafu.edu.cn (R.S.); fafulisiwei@163.com (S.L.); whyang@fafu.edu.cn (W.Y.); 1963zbq@163.com (B.Z.)
* Correspondence: fafuxsh@126.com

**Abstract:** The prime objective of biochar production is to contribute to nutrients recycling, reducing waste and converting useful bio-wastes into carbon rich products in the environment. The present study was intended to systematically evaluate the effect of pyrolysis conditions and characteristics of feedstock influencing the generation of biochar. The study revealed the nutritional importance of waste mushroom substrate (WMS) biochar which may elevate soil nutritional status and soil quality. The results showed that the yields and properties of WMS biochar depended principally on the applied temperature where pyrolysis at higher temperatures, that is, 600 °C and 700 °C produced biochar having high ash, P and K contents. Moreover, numerous useful macro and micro nutrients such as Ca, Mg, Fe and Zn were observed to positively correlate with the increase in temperature. The WMS biochar in our study is highly alkaline which can be used to rectify acidic soil pH. Overall our results suggest that WMS biochar being a rich source of nutrients can be the best remedy to maintain and further enhance the soil nutritional status. Thus by interpreting biochar feedstock characteristics and pyrolysis conditions, the regulation of tailored WMS biochar manufacturing and application in soil can be facilitated.

**Keywords:** FTIR; aromatic structure; atomic absorption spectroscopy; biochar nutrients

## 1. Introduction

The production of biochar has emerged as one of the most sustainable approaches to assist the issues regarding soil fertility, especially in countries like China where ample of biological waste can be utilized to generate biochar [1,2]. Biochar is a carbonaceous resultant product obtained from biomass of some plants or animals pyrolyzed at high temperature under anoxic or hypoxic conditions [3–5]. There are numerous ways in which biochar can potentially improve the soil quality. These include optimizing water-holding capacity and cation exchange capacity (CEC) as well as reducing the susceptibility of soil to erosion [6,7]. Furthermore, biochar can improve soil fertility through enhancing the availability of essential nutrients such as nitrogen (N), carbon (C) and phosphorus (P) [8,9] and reducing bioavailability of heavy metals [10] as well as acting as a source of carbon sequestration in soil [11]. However, pyrolysis conditions and feedstock types are crucial determinants of biochar properties [12]. With gradual increase in pyrolysis temperature polysaccharides constituents of the biomass are transformed to aromatic C structures characteristic of biochar [13,14] which contributes to their stability in soils [15]. Biochar obtained from pyrolysis of hard woods tends to have high N contents even when provided with higher temperature as compared to soft wood plants [1]. Biochar produced at low temperature is usually acidic in nature while a higher temperature tends to make it alkaline [16]. Moreover, a high

temperature intensifies various nutrients in biochar as well as increasing biochar properties in terms of its C percentage [17]. Feedstock is another critical factor to impact biochar characteristics. For instance, biochar produced from municipal wastes and poultry litter have lower aromatic C contents but higher ash contents than those produced from woody materials [18]. Biochar derived from animal and poultry manure is characterized by high nutritional contents, such as chicken manure contains high N, P, Ca and micronutrient contents, while coffee husk exhibits high K concentration [19]. Biochar obtained after pyrolysis of sugarcane bagasse and wood wastes have low nutrients but high lignin and cellulose contents [20,21].

Over the last 30 years, China has been making new breakthroughs and achieving domestic developments to create a revolution in food production industry. Mushroom industry expanded over more than 30 provinces in China, contributes a large amount of waste mushroom substrate (WMS) which may pile up and produce unhygienic conditions in the environment. Hence, the heaps of WMS need to be recycled or dumped properly in an efficient way to get the optimum useful product from them. Keeping in view the above discussion, the present study was undertaken to examine the effect of pyrolysis conditions on biochar derived from WMS and to evaluate what extent of nutrient concentration in WMS biochar can be expected. The waste mushroom substrate of species *Pleurotus eryngii*, commonly known as oyster mushroom, has been used as a feedstock to produce biochar using variable pyrolysis temperatures and times. We hypothesized that being rich in nutrients, WMS biochar can be a potential source to overcome nutrients dearth in soil. Many previous studies regarding biochar pyrolysis and characterization reported the influence of pyrolysis temperature on biochar properties but little are known about the effect of pyrolysis time on biochar properties. Therefore, the present study was proposed (1) to investigate the effects of the pyrolysis temperatures and times on the yield and physicochemical properties of biochar produced from WMS feedstock. (2) To evaluate WMS biochar nutritional values (N, P, K, Ca, Mg contents) and suggest guidelines for future research studies in biochar and its impact on soil fertility and environment.

It was expected that this research trial would provide useful data to reveal optimal pyrolysis conditions to maximize the beneficial use of WMS biochar as soil amendment. It was also expected that the study would help us to develop the best approach to properly manage the heaps of biological wastes by recycling them into the environment.

## 2. Materials and Methods

### 2.1. Feedstock Collection

The WMS was used as a feedstock for biochar production, collected from Gutian, which is the most famous county for producing edible fungus in China. The mushroom substrate selected to pyrolysis belonged to species *Pleurotus eryngii*. WMS feedstock was air dried and meshed into small pieces manually. The feedstock was then tested to evaluate primary contents in it (Table 1).

**Table 1.** Determination of chemical properties in waste mushroom substrate (WMS) feedstock.

| Property | Value/Unit | Property | Value/Unit | Property | Value/Unit |
|----------|-----------|----------|-----------|----------|-----------|
| pH | 6.34 | N | 2.18% | Cr | 5.581 (mg kg$^{-1}$) |
| CEC | 12 (cmol kg$^{-1}$) | Ca | 33.48 (g kg$^{-1}$) | Cu | 12.64 (mg kg$^{-1}$) |
| Total P | 3.72 (g kg$^{-1}$) | Mg | 6.57 (g kg$^{-1}$) | Zn | 96.58 (mg kg$^{-1}$) |
| Total K | 0.68 % | Pb | 25.26 (mg kg$^{-1}$) | Tl | 0.076 (mg kg$^{-1}$) |
| C % | 35.69% | Ni | 5.36 (mg kg$^{-1}$) | Mn | 1108.90 (mg kg$^{-1}$) |
| – | – | – | – | Cd | 0.40 (mg kg$^{-1}$) |

### 2.2. Preparation of Biochar

The pyrolysis of WMS feedstock was carried out in a fixed stainless steel reactor. Nitrogen gas was fed from the top at a flow rate of 500 mL/min for about 5 min to produce oxygen free conditions

during pyrolysis. Samples were pyrolyzed at various temperatures, that is, 400 °C, 500 °C, 600 °C and 700 °C for 1.5 h, 2 h, 2.5 h and 3 h, respectively. After certain hours of pyrolysis for each biochar the power supply was turned off and the system was cooled over night to reach the room temperature. The following day, biochar sample was removed, bagged, weighed and labeled for further analysis.

### 2.3. Determination of Biochar Properties

WMS biochar prepared under different pyrolysis conditions were then analyzed for various chemical and nutritional properties.

Briefly, 1.0 g biochar sample was combusted at 600 °C ± 20 °C for 4 h in a muffle furnace whereas the percentage of residual mass was calculated as the ash content. The pH (biochar: water; 1:20 suspension ratio) was measured by the standard method suggested by McLean (1982) using a portable pH meter (INESA Scientific Instrument Co., Ltd., Shanghai, China) [22]. Cation exchange capacity (CEC) was measured by using ammonium acetate solution using a flame photometer [23]. To determine mineral nutrients (N, P, K, Ca, Mg, Fe, Mn) and heavy metals (Cu, Zn, Pb, Cr, Cd, Ni, Tl) in biochar, WMS biochar samples were ground, sieved, weighed and put into triangular glass bottles. The weighed WMS biochar samples were then added with 5 mL concentrated nitric acid, covered with small funnel and left overnight. These samples were then digested in digestion blocks at 400 °C using a few drops of chloric acid at regular intervals until colorless transparent solutions appeared in the glass bottles. The transparent solutions were then used to make the respective dilutions for determination of mineral and heavy metals contents in WMS biochar using different instruments. The total Phosphorous (TP) was measured by molybdenum blue on a spectrophotometer (BioTek, Epoch2, Winooski, Vermont, USA) against the absorbance wavelength of 880 nm. Total K was measured using flame photometer (Wolf, 1982). Atomic absorption spectroscopy (AAS) was used to examine iron (Fe), calcium (Ca) and magnesium (Mg) following standard protocol by Miyazawa (1984) whereas ICP-MS was used to determine heavy metals (Cu, Zn, Pb, Cr, Cd, Mn, Ni, Tl) concentrations in biochar (EPA 3052 methods).

The surface area of WMS biochar samples was measured using Brunauer–Emmett–Teller (BET) and its respective application (Micromeritics GeminiV). C percentage (%), N (%) and C: N ratio was determined using elemental analyzer (LECO Corporation, St. Joseph, MI, USA).

### 2.4. Fourier Transform Infrared Spectroscopy Analysis

To perform Fourier transform infrared spectroscopy (FTIR), a small amount of WMS biochar sample and dry KBr in a ratio of 1:100 or 1:200 were ground together in an agate mortar. After being ground for 5 min the greyish white powder was pressed downward to form a delicate frame which is used for laser scanning using FTIR spectrometer (Bruker vertex 70) to determine various aromatic groups in biochar samples at different pyrolysis conditions. FTIR spectra for different WMS biochar samples and feedstock were recorded with a resolution of 4 cm$^{-1}$ in wavelength ranging between 500 and 4000 cm$^{-1}$.

### 2.5. Statistical Analysis

Data are expressed as means ± standard errors of triplicate measurements. All data was analyzed using one-way ANOVA and applying Duncan's test at a 95% confidence interval ($p < 0.05$). IBM SPSS Statistics 20.0 was used to determine the significance levels among various WMS biochar obtained at different pyrolysis conditions. For surface area application of BET and t-plot analysis software available with instrument (Micromeritics GeminiV) was used. Software Omnic 8.0 was used to analyze FTIR spectra for biochar samples. Origin pro 8.5 was used to determine FTIR peaks and draw graphs.

## 3. Results

### 3.1. Biochar Yield

The effect of pyrolysis temperature on the WMS biochar yield was shown in Figure 1. The increase of pyrolysis temperature led to a decrease in the yields of WMS biochar. A decrease of 8.08% and 33.7% in WMS biochar yield was found when temperature increased from 400 °C to 700 °C, respectively. The effect of pyrolysis time on biochar yield also produced similar results (Figure 1). At 400 °C, yield decreased for only 6.8% when time was increased from 1.5 h to 2 h but yield decrease was more when it was raised up to 2.5 and 3 h. Results also indicated that pyrolysis time affected biochar yield at lower and higher temperatures more than moderate temperatures such as at 500 °C and 600 °C, the yield decrease was 16.98% and 10.41% respectively while at 400 °C and 700 °C, yield decreased for 21.8% and 25.6% with time increase from 1.5 to 3 h, respectively.

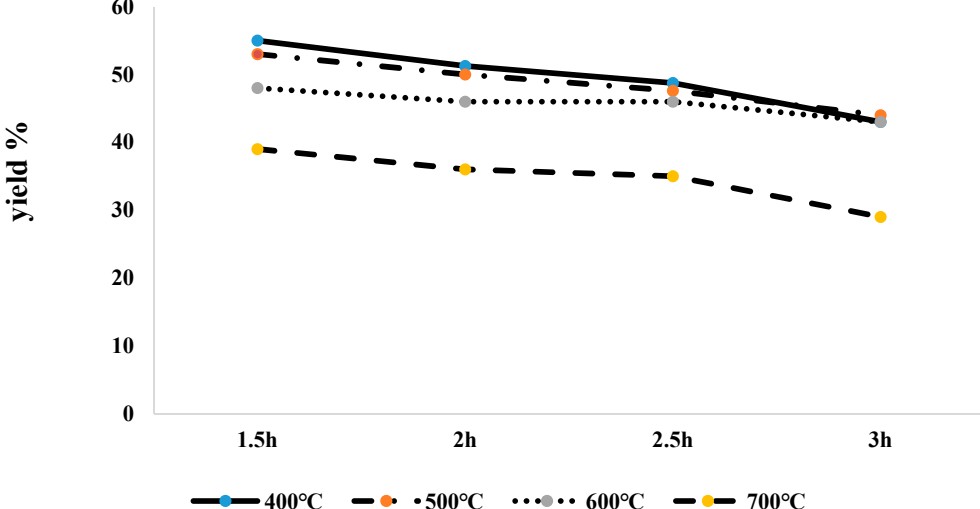

**Figure 1.** Yield of WMS biochar produced at different pyrolysis temperatures and times.

### 3.2. pH and Ash Contents of WMS Biochar

The pyrolysis temperature proved to have a significant positive correlation with ash and pH. Initially, an increasing trend in pH from lower to higher temperatures was observed. Ash contents were found to be the highest when WMS biochar was pyrolyzed at 700 °C. Ash contents ranged from 54.34% to 32.33% at temperature 700 °C to 400 °C, respectively (Table 2). The pH values of WMS biochar obtained under different pyrolysis temperatures and times were given in Tables 2 and 3, respectively. The highest pH value recorded for the WMS biochar was pH 10.63 (at 700 °C) for 3 h pyrolysis time. The lowest value of pH was observed for WMS biochar pyrolyzed at 400 °C, that is, 8.66. The results indicated that pyrolysis temperature had dominating effect on ash contents and pH of WMS biochar than pyrolysis duration (Tables 2 and 3).

**Table 2.** Effect of different pyrolysis temperatures on ash contents, pH, CEC and BET surface area of WMS biochar.

| Temperature (°C) | Ash (%) | pH | CEC (cmol kg$^{-1}$) | Surface Area (m$^2$g$^{-1}$) |
|---|---|---|---|---|
| 400 | 32.33 ± 0.79 d | 8.95 ± 0.32 c | 32.24 ± 0.95 a | 3.36 ± 0.24 a |
| 500 | 41.08 ± 1.31 c | 9.24 ± 0.15 b | 31.10 ± 1.72 ab | 2.72 ± 0.13 b |
| 600 | 47.08 ± 0.97 b | 9.40 ± 0.18 b | 28.24 ± 1.42 ab | 1.97 ± 1.77 c |
| 700 | 54.34 ± 0.81 a | 10.28 ± 0.17 a | 22.70 ± 1.73 b | 1.71 ± 1.49 c |

Readings are means of three treatment replicates. Means followed by the same letter within a line are not significantly different at the 5% level.

**Table 3.** Effect of different pyrolysis times on ash contents, pH, CEC and BET surface area of WMS biochar.

| Temperature (°C) | Time (h) | Ash (%) | pH | CEC (cmol kg$^{-1}$) | Surface Area (m$^2$g$^{-1}$) |
|---|---|---|---|---|---|
| 400 | 1.5 | 28.67 ± 0.88 b | 8.66 ± 0.13 c | 29.53 ± 0.39 a | 3.86 ± 0.49 a |
| | 2 | 32.67 ± 1.2 a | 8.67 ± 0.13 c | 31.90 ± 0.97 a | 4.04 ± 0.37 a |
| | 2.5 | 33.33 ± 0.88 a | 9.10 ± 0.21 b | 33.80 ± 3.36 a | 2.95 ± 0.11 ab |
| | 3 | 34.67 ± 0.88 a | 9.38 ± 0.04 a | 34.56 ± 0.29 a | 2.60 ± 0.32 b |
| 500 | 1.5 | 35.67 ± 0.88 b | 8.57 ± 0.06 c | 29.28 ± 0.92 b | 2.65 ± 0.37 a |
| | 2 | 38.67 ± 0.88 b | 8.72 ± 0.17 b | 32.40 ± 4.16 a | 2.88 ± 0.24 a |
| | 2.5 | 43.67 ± 1.2 a | 9.78 ± 0.16 b | 32.25 ± 0.25 a | 2.72 ± 0.28 a |
| | 3 | 46.33 ± 0.67 a | 9.89 ± 0.17 a | 30.5 ± 0.39 ab | 2.57 ± 0.26 a |
| 600 | 1.5 | 44.33 ± 1.33 b | 8.69 ± 0.11 d | 30.0 ± 0.64 a | 2.45 ± 0.39 a |
| | 2 | 45.33 ± 0.33 b | 9.10 ± 0.03 c | 26.57 ± 0.53 b | 2.27 ± 0.25 a |
| | 2.5 | 47.33 ± 0.67 ab | 9.53 ± 0.037 b | 25.69 ± 0.29 b | 1.95 ± 0.42 ab |
| | 3 | 51.33 ± 2.02 a | 10.26 ± 0.12 a | 25.39 ± 0.26 b | 1.19 ± 0.78 b |
| 700 | 1.5 | 53.00 ± 1.76 a | 9.37 ± 0.16 b | 25.15 ± 0.67 a | 2.04 ± 0.24 a |
| | 2 | 53.67 ± 1.76 a | 10.52 ± 0.19 a | 22.14 ± 0.148 b | 2.00 ± 0.16 a |
| | 2.5 | 53.37 ± 1.88 a | 10.60 ± 0.20 a | 21.84 ± 0.256 b | 1.74 ± 0.28 ab |
| | 3 | 55.33 ± 1.86 a | 10.63 ± 0.15 a | 21.69 ± 0.148 b | 1.05 ± 0.12 b |

Readings are means of treatment triplicates. Means followed by the same letter within a line are not significantly different at the 5% level.

### 3.3. CEC and BET Surface Area of WMS Biochar

An inverse relationship between pyrolysis temperature and CEC of WMS biochar was predicted. In our study, CEC of WMS biochar decreased from 32.24 cmolc kg$^{-1}$ to 22.70 cmolc kg$^{-1}$ as temperature changed from 400 °C to 700 °C, respectively, whereas pyrolysis time did not show any significant influence on CEC of WMS biochar (Tables 2 and 3). WMS biochar CEC values varied greatly and were mainly dependent on the biomass and temperature used in the pyrolysis process (Table 2). In our study, biochar produced at higher temperature showed the lower values of CEC (Table 2) whereas WMS biochar obtained at lower temperatures (400 °C and 500 °C) showed higher CEC values.

The effect of pyrolysis temperature and time on the BET surface of the WMS biochar is shown in Tables 2 and 3. When the pyrolysis was carried out at 400 °C for 2 and 1.5 h, the values of BET surface area were maximum, that is, 4.04 m$^2$g$^{-1}$ and 3.86 m$^2$g$^{-1}$ respectively. However, at increasing pyrolysis temperature and time, the trend was reversed. In case of BET surface area of WMS biochar, both time and temperatures seemed affecting the values of surface area. At 400 °C, as time increases from 2 to 2.5 and 3 h, there is a slight decrease in surface area of WMS biochar. Similar trend was observed at high pyrolysis temperatures, that is, when time was increased up to a certain point, surface area of biochar was reduced (Table 3).

### 3.4. C, N Contents of WMS Biochar

Pyrolysis temperature exhibited a paramount effect on biochar elemental compositions rather than pyrolysis time (Table 4). The statistical analysis of data reveals that resultant biochar C contents are significantly correlated to pyrolysis temperature. With an increase in the pyrolysis temperature from 400 to 700 °C, C contents of WMS biochar increased from 36.46% to 41.08 wt %. In general, high N contents of biochar can provide nutrients to soil and improve crop productivity. Therefore N% was determined using an elemental analyzer. N contents decreased from 2.36 wt % to 1.17% whereas as the pyrolysis temperature rose, C/N ratios gradually increased (Table 4). Both pyrolysis temperature and time seemed affecting N contents and C/N ratio of WMS biochar (Tables 4 and 5).

**Table 4.** Effect of different pyrolysis temperatures on Total P, K, C, N contents and C:N of WMS biochar.

| Temperature (°C) | C (%) | N (%) | C:N | Total P (g kg$^{-1}$) | Total K (%) |
|---|---|---|---|---|---|
| 400 | 36.46 ± 0.42 d | 2.36 ± 0.08 a | 15.68 ± 0.64 d | 18.00 ± 0.56 a | 1.97 ± 0.09 d |
| 500 | 38.46 ± 0.13 c | 1.76 ± 0.06 b | 22.10 ± 0.79 c | 18.01 ± 0.82 a | 2.23 ± 0.05 c |
| 600 | 39.98 ± 0.08 b | 1.28 ± 0.01 c | 32.55 ± 0.49 b | 19.70 ± 1.25 a | 2.39 ± 0.08 b |
| 700 | 41.08 ± 0.16 a | 1.17 ± 0.01 c | 35.11 ± 0.91 a | 19.75 ± 1.05 a | 2.53 ± 0.07 a |

Values are means of three replicates of each treatment ± SE using $\alpha$ = 0.05.

**Table 5.** Effect of different pyrolysis times on Total P, K, C, N contents and C:N of WMS biochar.

| Temperature (°C) | Time (h) | C (%) | N (%) | C:N | Total P (g kg$^{-1}$) | Total K (%) |
|---|---|---|---|---|---|---|
| 400 | 1.5 | 34.55 ± 0.11 c | 2.58 ± 0.26 a | 13.71 ± 1.54 b | 16.83 ± 0.46 bc | 1.37 ± 0.038 b |
| | 2 | 36.23 ± 0.35 b | 2.46 ± 0.01 a | 14.73 ± 0.13 b | 18.09 ± 0.50 ab | 2.14 ± 0.058 a |
| | 2 | 37.22 ± 0.60 ab | 2.30 ± 0.08 a | 16.22 ± 0.86 ab | 18.32 ± 2.63 ab | 2.16 ± 0.038 a |
| | 3 | 37.83 ± 0.51 a | 2.09 ± 0.06 a | 18.07 ± 0.76 a | 18.78 ± 0.69 ab | 2.23 ± 0.057 a |
| 500 | 1.50 | 37.99 ± 0.11 b | 2.01 ± 0.02 a | 18.91 ± 0.22 b | 16.95 ± 1.59 bc | 2.07 ± 0.019 c |
| | 2 | 38.17 ± 0.05 ab | 1.87 ± 0.06 b | 20.48 ± 0.76 b | 18.90 ± 1.13 ab | 2.10 ± 0.038 c |
| | 2.5 | 38.71 ± 0.02 a | 1.61 ± 0.03 c | 23.93 ± 0.36 a | 18.21 ± 0.40 ab | 2.28 ± 0.019 b |
| | 3 | 38.60 ± 0.32 a | 1.56 ± 0.03 c | 25.09 ± 0.63 a | 18.32 ± 0.19 ab | 2.49 ± 0.057 a |
| 600 | 1.50 | 39.59 ± 0.03 b | 1.32 ± 0.01 a | 29.91 ± 0.13 b | 18.54 ± 1.79 bc | 2.10 ± 0.03 c |
| | 2 | 39.98 ± 0.01 ab | 1.35 ± 0.02 a | 32.92 ± 0.71 a | 19.02 ± 1.09 ab | 2.18 ± 0.03 c |
| | 2.5 | 40.13 ± 0.11 a | 1.25 ± 0.02 b | 33.54 ± 0.09 a | 19.70 ± 0.59 ab | 2.51 ± 0.019 b |
| | 3 | 40.22 ± 0.11 a | 1.21 ± 0.02 b | 33.84 ± 0.15 a | 21.55 ± 0.64 a | 2.78 ± 0.00 a |
| 700 | 1.50 | 40.76 ± 0.13 b | 1.20 ± 0.01 a | 25.14 ± 0.62 c | 16.75 ± 1.70 bc | 2.12 ± 0.019 b |
| | 2 | 40.67 ± 0.45 ab | 1.18 ± 0.02 a | 29.05 ± 0.10 b | 19.40 ± 3.018 ab | 2.51 ± 0.07 b |
| | 2.5 | 41.16 ± 0.19 a | 1.16 ± 0.05 b | 30.19 ± 0.54 b | 20.51 ± 0.64 ab | 2.58 ± 0.019 a |
| | 3 | 41.74 ± 0.14 a | 1.12 ± 0.01 c | 33.38 ± 0.51 a | 22.35 ± 1.28 a | 2.92 ± 0.019 a |

Values are means of three replicates of each treatment ± SE using $\alpha$ = 0.05.

## 3.5. Nutritional Analysis

Concentrations of P and K were increased slightly with the rise in temperature. The difference in P contents at low and high pyrolysis temperature was non-significant though a significant difference was observed in P contents at various pyrolysis times (Tables 4 and 5). In case of K, both pyrolysis temperature and time had a significant effect on K concentration of WMS biochar and K% increased from 1.97% to 2.53% when temperature was raised from 400 °C to 700 °C. Ca and Mg concentrations also showed an increase with the rising pyrolysis temperature and these nutrients were the highest in WMS biochar obtained at 700 °C. As compared to nutrient content in feedstock, the total contents of Ca and Mg rose to 74% and 6% respectively, when pyrolysis process was conducted at a temperature of 700 °C. An increase in Fe and Mn contents was also observed with the increase in pyrolysis temperature and times (Tables 6 and 7).

**Table 6.** Effect of different pyrolysis temperatures on Fe, Ca and Mg contents of WMS biochar.

| Temperature (°C) | Ca (g kg$^{-1}$) | Mg (g kg$^{-1}$) | Fe (g kg$^{-1}$) | Mn (mg kg$^{-1}$) |
|---|---|---|---|---|
| 400 | 98.65 ± 8.38 a | 5.16 ± 0.41 b | 25.36 ± 1.76 b | 586.07 ± 78.18 a |
| 500 | 105.67 ± 7.98 a | 5.19 ± 0.15 b | 25.41 ± 1.30 b | 643.11 ± 65.11 a |
| 600 | 115.47 ± 3.14 a | 5.81 ± 0.15 ab | 26.25 ± 2.15 b | 682.53 ± 28.25 a |
| 700 | 117.47 ± 3.10 a | 6.61 ± 0.13 a | 31.97 ± 1.96 a | 705.6 ± 19.312 a |

Readings are means of treatment triplicates. Means followed by the same letter within a row are not significantly different at the 5% level.

**Table 7.** Effect of different pyrolysis times on Fe, Ca and Mg contents of WMS biochar.

| Temperature (°C) | Time (h) | Ca (g kg$^{-1}$) | Mg (g kg$^{-1}$) | Fe (g kg$^{-1}$) | Mn (mg kg$^{-1}$) |
|---|---|---|---|---|---|
| 400 | 1.5 | 72.038 ± 30.17 a | 4.08 ± 1.58 b | 20.50 ± 0.20 b | 528.59 ± 22.27 b |
| | 2 | 101.68 ± 1.58 a | 5.09 ± 0.08 b | 24.80 ± 4.16 ab | 570.3 ± 8.679 ab |
| | 2.5 | 102.90 ± 2.63 a | 5.10 ± 0.22 a | 23.26 ± 2.46 b | 598.09 ± 22.18 a |
| | 3 | 117.93 ± 8.44 a | 5.68 ± 0.52 a | 32.85 ± 1.70 a | 627.23 ± 212.7 a |
| 500 | 1.5 | 81.65 ± 7.43 c | 4.64 ± 0.07 b | 19.8 ± 0.58 b | 614.82 ± 15.58 a |
| | 2 | 97.58 ± 22.1 bc | 5.08 ± 0.25 b | 25.48 ± 1.80 a | 624.49 ± 7.32 a |
| | 2.5 | 119.39 ± 3.24 ab | 5.78 ± 0.086 a | 26.24 ± 0.20 a | 658.6 ± 22.49 a |
| | 3 | 124.09 ± 1.35 a | 5.95 ± 0.019 a | 30.12 ± 2.80 a | 681.46 ± 45.07 a |
| 600 | 1.5 | 102.70 ± 9.19 b | 5.24 ± 0.18 b | 19.38 ± 0.68 c | 662.26 ± 20 b |
| | 2 | 115.84 ± 1.29 ab | 5.64 ± 0.37 ab | 22.346 ± 0.58 bc | 684.32 ± 8.5 a |
| | 2.5 | 119.19 ± 2.29 a | 6.18 ± 0.09 a | 26.7 ± 3.47 b | 690.65 ± 41.2 a |
| | 3 | 124.09 ± 1.35 a | 6.20 ± 0.054 a | 36.56 ± 2.19 a | 692.91 ± 46.5 a |
| 700 | 1.5 | 102.70 ± 9.19 b | 6.19 ± 0.108 c | 31.12 ± 4.60 a | 621.67 ± 9.69 d |
| | 2 | 118.24 ± 1.23 ab | 6.44 ± 0.21 bc | 34.97 ± 0.75 a | 681.5 ± 12.54 c |
| | 2.5 | 121.29 ± 2.26 ab | 6.93 ± 0.048 ab | 30.26 ± 5.00 a | 726.25 ± 9.9 b |
| | 3 | 130.16 ± 1.81 a | 7.02 ± 0.24 a | 31.55 ± 5.50 a | 792.97 ± 3.65 a |

Readings are means of treatment triplicates. Means followed by the same letter within a row are not significantly different at the 5% level.

### 3.6. Heavy Metal Analysis

The results revealed that concentrations of Zn, Cu and Ni increased in WMS biochar samples as compared to the feedstock (Tables 1 and 8), while Cr, Pb and Tl contents were decreased in biochar after pyrolysis (Tables 8 and 9). Pyrolysis temperature played a paramount role in determining the properties of heavy metals contents in WMS biochar. It also showed that Pb and Tl in WMS biochar were significantly lower than WMS feedstock which made WMS biochar safer to apply in soil.

**Table 8.** Effect of different pyrolysis temperatures on heavy metal contents of WMS biochar.

| Temperature (°C) | Cu (mg kg$^{-1}$) | Zn (mg kg$^{-1}$) | Cr (mg kg$^{-1}$) | Ni (mg kg$^{-1}$) | Pb | Tl |
|---|---|---|---|---|---|---|
| 400 | 21.74 ± 3.71 b | 150.32 ± 27.65 d | 4.33 ± 0.57 b | 6.15 ± 0.73 b | | |
| 500 | 24.21 ± 1.59 ab | 173.8 ± 26.18 c | 5.24 ± 0.96 ab | 8.33 ± 1.32 ab | Below detection limit | Below detection limit |
| 600 | 24.76 ± 4.28 ab | 222.27 ± 47.62 b | 8.70 ± 1.30 a | 9.05 ± 2.00 ab | | |
| 700 | 29.52 ± 1.20 a | 265.79 ± 10.33 a | 9.84 ± 1.120 a | 11.70 ± 1.66 a | | |

Readings are means of treatment triplicates. Means followed by the same letter within a line are not significantly different at the 5% level.

**Table 9.** Effect of different pyrolysis times on heavy metal contents of WMS biochar.

| Temperature (°C) | Time (h) | Cu (mg kg$^{-1}$) | Zn (mg kg$^{-1}$) | Cr (mg kg$^{-1}$) | Ni (mg kg$^{-1}$) | Pb | Tl |
|---|---|---|---|---|---|---|---|
| 400 | 1.5 | 19.49 ± 3.69 a | 108.8 ± 23.41 b | 3.63 ± 0.27 a | 4.41 ± 0.25 b | | |
| | 2 | 22.23 ± 5.77 a | 135.9 ± 4.3 ab | 3.18 ± 0.85 a | 4.59 ± 1.47 ab | | |
| | 2.5 | 22.77 ± 10.7 a | 167.59 ± 9.8 ab | 4.37 ± 1.47 a | 6.89 ± 0.49 ab | | |
| | 3 | 22.47 ± 1.27 a | 191 ± 20.48 a | 6.16 ± 1.27 a | 8.68 ± 1.82 a | | |
| 500 | 1.5 | 19.82 ± 1.449 b | 146.33 ± 23.14 b | 3.64 ± 0.91 a | 4.16 ± 1.08 c | | |
| | 2 | 21.23 ± 0.316 b | 164.82 ± 9.38 ab | 4.94 ± 1.50 a | 5.44 ± 0.134 c | | |
| | 2.5 | 26.93 ± 1.62 ab | 169.5 ± 17.14 ab | 5.89 ± 0.88 a | 8.78 ± 1.63 b | Below detection limit | Below detection limit |
| | 3 | 28.86 ± 4.33 a | 211.73 ± 10.7 a | 6.49 ± 3.75 a | 14.92 ± 0.26 a | | |
| 600 | 1.5 | 21.84 ± 0.42 c | 148.00 ± 15 ab | 4.56 ± 0.70 c | 6.33 ± 2.38 c | | |
| | 2 | 23.23 ± 15.84 ab | 217.9 ± 12.08 ab | 5.11 ± 1.69 c | 6.37 ± 0.07 c | | |
| | 2.5 | 24.96 ± 0.547 ab | 219.7 ± 15.14 ab | 12.50 ± 7.31 b | 7.48 ± 3.96 b | | |
| | 3 | 29.03 ± 4.422 a | 305.35 ± 17.10 a | 13.09 ± 2.48 a | 16.03 ± 5.82 a | | |
| 700 | 1.5 | 28.46 ± 2.15 b | 153.7 ± 21.43 c | 6.44 ± 1.96 b | 7.18 ± 0.49 b | | |
| | 2 | 29.18 ± 1.09 ab | 230.9 ± 18.8 bc | 6.55 ± 2.22 b | 9.25 ± 1.51 b | | |
| | 2.5 | 29.68 ± 1.84 ab | 338.4 ± 2.238 ab | 13.20 ± 1.92 a | 11.92 ± 1.74 ab | | |
| | 3 | 30.79 ± 0.47 a | 341.09 ± 7.676 a | 13.23 ± 1.98 a | 18.48 ± 4.37 a | | |

Readings are means of treatment triplicates. Means followed by the same letter within a line are not significantly different at the 5% level.

*3.7. FTIR Analysis*

The FTIR analyses of WMS biochar at different pyrolysis conditions indicate significant contrasting results (Figures 2–5). The aromatic structure (C=C stretch) in FTIR of WMS biochar gradually lost its strength as the temperature rose up to 700 °C. Both temperature and pyrolysis time seemed affecting the FTIR spectrum of WMS biochar. Overall, aliphatic structure predominated only at low temperatures, that is, 400 °C and 500 °C and the strength of aliphatic peaks (3100–3600 cm$^{-1}$) was observed to diminish with the increase in pyrolysis temperature.

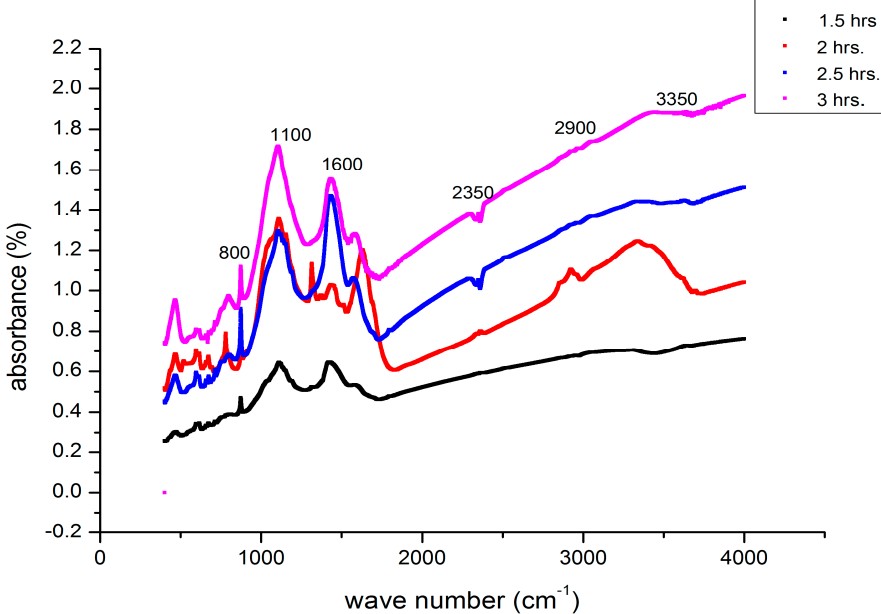

**Figure 2.** Fourier transform infrared (FTIR) spectra of WMS biochar obtained at 400 °C pyrolysis temperature and various times

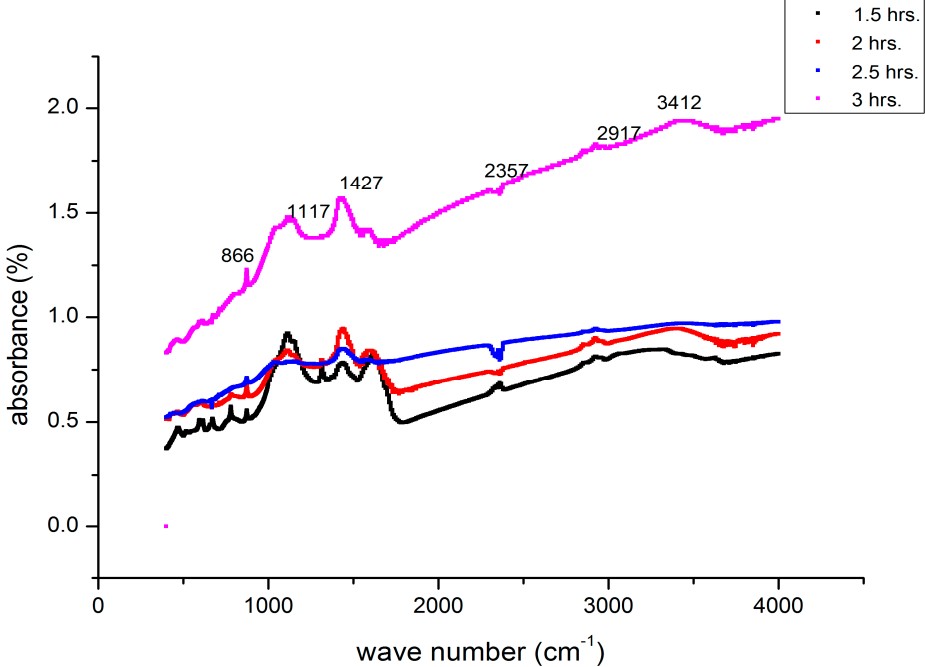

**Figure 3.** FTIR spectra of WMS biochar obtained at 500 °C pyrolysis temperature and various times.

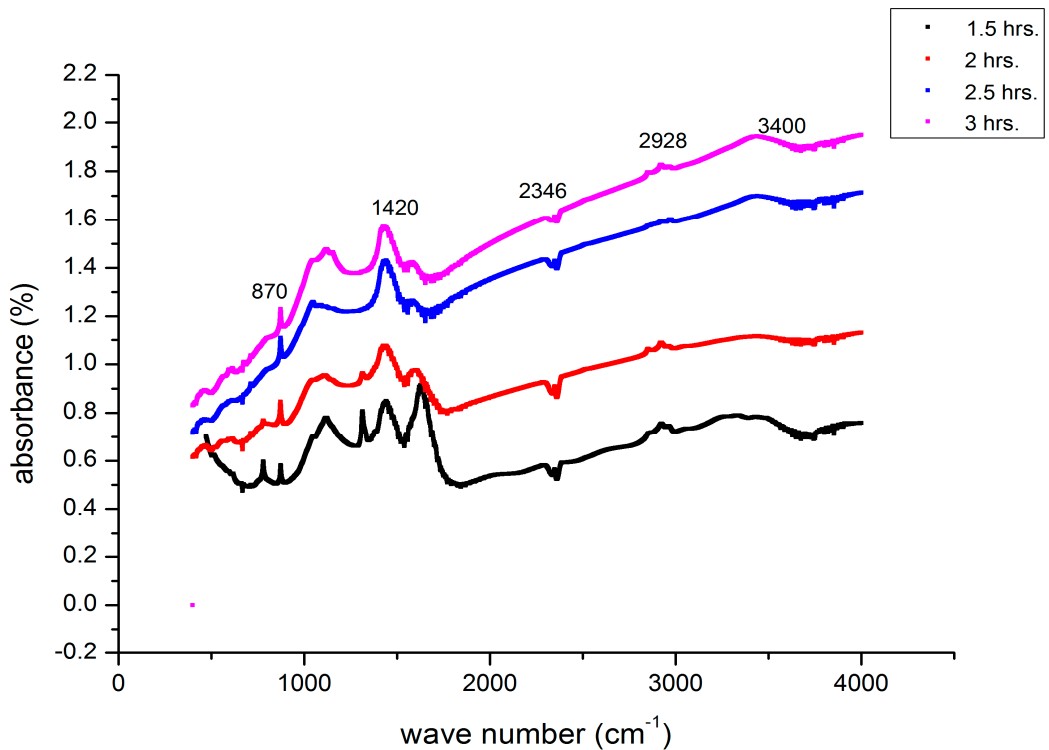

**Figure 4.** FTIR spectra of WMS biochar obtained at 600 °C pyrolysis temperature and various times.

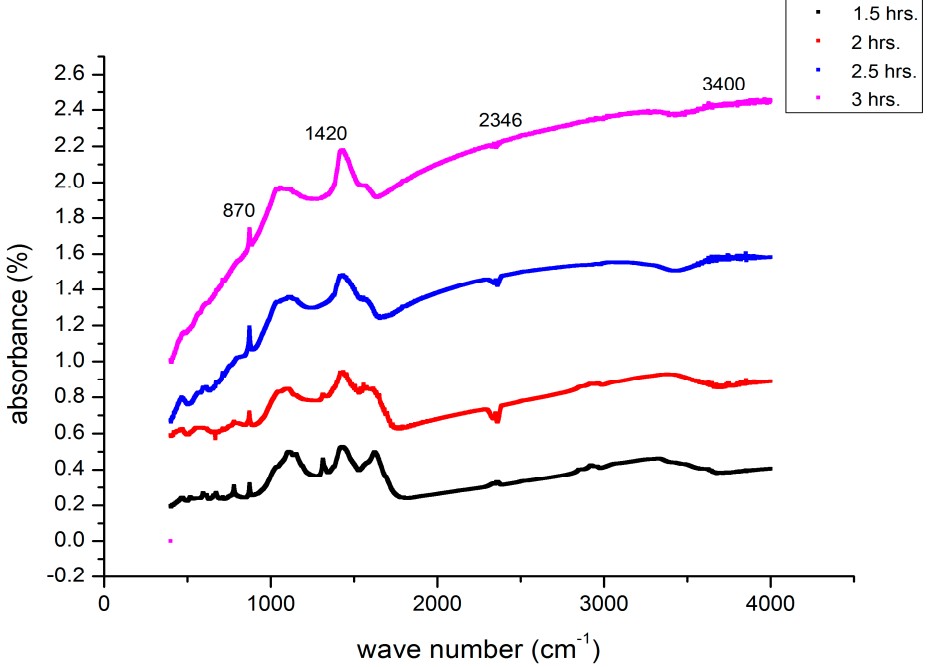

**Figure 5.** FTIR spectra of WMS biochar obtained at 700 °C pyrolysis temperature and various times.

## 4. Discussion

### 4.1. Physicochemical Properties of WMS Biochar under Various Pyrolysis Temperatures and Times

The significant decrease in biochar yield with the increase in pyrolysis temperature is reported in the literature, as in the case of oak, pine, sugarcane and peanut shell biochar [8]. The decrease in biochar yields with increasing temperature could be due to deterioration of the basic primary structure

of biochar after pyrolysis process. At low pyrolysis temperature, biochar was partially combusted resulting in higher yield while high temperature caused biochar material to combust completely consequently producing lower yield [4,7].

In the present research, ash contents of WMS biochar were observed to be higher than those of safflower seed cake, sugarcane straw, rice hull, poultry litter and sawdust reported previously [4,24]. Whereas, pyrolysis time did not have a significant effect on pH but the pH of the WMS biochar increased significantly with increasing pyrolysis temperatures. The pH of the WMS biochar ranged from 8.66 to 10.63 which was similar to values reported for biochar produced from sugar beet tailings, sewage sludge and sugarcane bagasse at high pyrolysis temperature [25–27] (Tables 2 and 3). An increase in biochar pH with increasing pyrolysis temperature has been reported for corn straw, pine, poultry litter, sewage sludge and sugarcane straw biochar [13,28–31]. The effect of pyrolysis temperature on pH mainly relates to two aspects: (1) with increasing temperature, there is an enrichment of basic cations in the ashes, which may be associated with alkaline species, such as carbonates, oxides and hydroxides [28,32] and (2) a reduction in the concentration of acidic surface functional groups [33]. The nature of the aromatic C structures associated with acidic functional groups can affect biochar CEC and adsorption capacity [34]. Decrease in CEC of biochar with increased pyrolysis temperature can be explained by degradation in volatile organic compounds and acid functional groups ($-COO^-$ and $-O^-$) [29,35] which have been associated to the negative surface charge of biochar [28,36]. At pyrolysis temperature 400 °C, the number of micropores significantly increased with the removal of volatile matter, giving rise to an increase in the pore volume and surface area. At high temperatures, modification in structural order, pore widening and merging of neighboring pores seem to predominate, leading to a decrease in the surface area. Besides, pores in the biochar might be constricted as a result of squeezing, softening, fusion and carbonization [37].

## 4.2. Nutritional Properties of WMS Biochar under Various Pyrolysis Temperatures and Times

C and N cycles are the most important nutrients in soil which are profoundly affected by pyrolysis temperature of biochar. Considering these, it is expected a greater aromatic character for WMS biochar produced at higher temperature. The use of these biochar with a possible high residence time may be an important strategy to increase C sequestration in soils, acting to offset greenhouse gas emissions [38].

Our data revealed the obvious increase in C content of biochar with the increase in temperature but the loss of N contents was recorded. The C content of WMS biochar was slightly higher than that of straw and chicken manure biochar but less than that of conocarpus wastes, hard wood and safflower seed cake biochar, which have been reported previously. In case of N, WMS biochar had more N contents than hard wood (oak) and straw biochar [1,21,39,40]. It is implied that these properties corresponded to their feedstocks instead of pyrolysis temperature and time which determine their properties [30].

P and K contents in WMS biochar are higher than those of hard wood, sugarcane straw, rice husk and poultry litter, wheat and rice straw biochar [8,24,41] reported previously. Macro nutrients (P, K) of WMS biochar were greatly higher than animal and plants biochar feedstock which makes it a great substitute to add in soil [24]. The results of the present study showed that Ca, Mg, Mn and Fe concentrations increased with increasing pyrolysis temperature, which may be due to concentration of these elements in biochar samples with temperature. Additionally, these elements might not be lost by volatilization [35]. Hence, the saturation of alkaline elements could be responsible for inducting liming effects in biochar pyrolyzed at high temperatures [1,35]. Pb and Tl contents in WMS biochar were below the detection limit of ICP-MS which support the nutritional reliability of WMS biochar to apply in soil.

C contents (%) and biochar FTIR configurations can be used as predictors of C persistence in soils (Figures 2–5). High C content and FTIR spectrum features recorded for biochar derived from high temperatures are key indices of the aromatic character, stability against degradation in soils, and, consequently, high C residence time in biochar-treated soils. Generally, the effect of pyrolysis time on the biochar FTIR profiles is not very pronounced. The increasing upward drift in the primary baseline of the spectrum from low to high wave numbers at the high pyrolysis temperature was

probably due to the increased aromatics compounds [8,42]. An observed increase in intensity in the 1600 cm$^{-1}$ region with increasing pyrolysis temperature has been attributed to an increasing degree of saturation of aromatic organic contents [21]. Stronger C=O stretching peaks in 1600 cm$^{-1}$ region may also indicate the formation of ketones, esters, anhydrides and carboxylic C [4,43]. In case of C contents, WMS biochar produced at lower temperature exhibit intense FTIR peaks which can be readily degraded by microbes in soil enhancing the soil fertility, whereas WMS biochar obtained at high pyrolysis temperature can be good for soil C sequestration [18,44].

In our study, WMS biochar has the potential to increase soil acidity, buffering capacity and to neutralize soil acidity. High pH of WMS biochar indicates the presence of alkaline chemical species which can reduce Fe and Mn availability and increase soil CEC [28,31] which may decrease the precipitation and adsorption of P, as well as enhancing the supply of Ca and K to plants [45]. It is observed that P availability in biochar amended soil was initially lower than a similar manure amended soil but that the availability in the manure amended soil declined gradually whereas it increased in the biochar amended soil, in a long term experiment [46,47]. High immediate P availability, which can be an advantage for crops with an instantaneous need for P is thus achieved using relatively low pyrolysis temperatures. In our study, WMS biochar produced at low temperature can act as a good source of P in soil whereas high temperature WMS biochar can provide nutrients in soil in long terms. The high Ca and K contents in WMS biochar can significantly replace conventional sources of K and Ca, which suggests the high nutritious value of WMS biochar.

## 5. Conclusions

The results of pyrolysis study of WMS biochar indicate that pyrolysis temperature is significantly associated to biochar yield, pH, CEC and surface area and was found to be the most significant factor determining WMS biochar properties. Pyrolysis temperature displayed a positive relationship with pH, CEC, surface area and ash content. The surface area was significantly influenced by pyrolysis time, which is often overlooked in the literature. The biochar derived from utilized mushroom substrate can be used as a multi element supplement for soil and plants for being a highly valuable source of P, K, Ca, Mg and Fe. The presence of innocuous concentration of heavy metals also provides ideal conditions for application of WMS biochar in soil. WMS biochar mainly cannot be used as a C sequester in soil but it can be a great remedy to avoid N leaching in soil and increasing nutrients (P, K, Ca, Mg) availability in long terms. Furthermore, it is also concluded that biochar utilized for an application should match the aim of the use in soil application.

**Author Contributions:** R.S. conducted the experiments and wrote the original draft. S.L. analyzed the data; W.Y. reviewed and edited the draft; B.Z. provided insight and support for the research experiment. S.X. supervised the experiment project and approved the final version.

**Funding:** We are grateful to the Institute of Agriculture Ecology in Fujian Academy of Agriculture Sciences for project cooperation. This research was supported by both Special Projects of Science and Technology funded by Department of Science & Technology, Fujian (2017NZ0001) and Special Innovation Project of Science and Technology at Fujian Agriculture and Forestry University (KFA17397A).

**Conflicts of Interest:** The authors declare no conflict of interest.

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
