# Peer review of "Assessment of Physicochemical and Nutritional Characteristics of Waste Mushroom Substrate Biochar under Various Pyrolysis Temperatures and Times"

_sustainability, doi:10.3390/su11010277_

Round 1

Reviewer 1 Report

My first concern is the term used for mushroom compost. I realize they reference it being an issue in China, but I am not aware of anyone in the world that calls it "Industrial waste mushroom substrate." A term like this will upset many in the industry, at least in the North America and Europe. In Agaricus production in the United States "mushroom substrate" refers to the composted material mushrooms are grown on. At one time the industry was required to call the material removed from the house after the crop was finished as "spent mushroom substrate." In later years the Pennsylvania State government gave permission to call it "mushroom compost," because compost is what it is. This then makes it comparable to other types of compost.

The English name of the mushroom species and the materials used for the substrate should be given in both the abstract and in the text. In the English speaking world, when only the word "mushroom" is used as an identifier, it is assumed the author is speaking of Agaricus bisporus. Later the paper says the mushroom is H. E. Bigelow. This seems to be the name of the person who originally described the mushroom species. It is not a species of mushroom. The fact that it is italicized gives the impression that it is the scientific name of the mushroom.

Overall I am not sure the point of the paper. The authors begin the abstract by stating the benefits of producing biochar, but do not explain how producing it produces the benefits. The paper makes the claim that producing biochar increased the amount of carbon in the biochar. Carbon is not created. Its percentage of the overall mass of the material increases through the elimination of other materials through the heating process. The benefit of reduced carbon emissions would come via reduced transportation because there is less material to haul.

Another way I thought the paper may have went was to tell me what the optimal temperature and time for processing would be to optimize biochar quality. It stated some obvious facts like as the process used higher temperatures and longer processing times, there was less material and a higher percentage of inorganic compounds, such as ash and heavy metals, but what levels of these different factors do we want? If it cited a publication that defined the best biochar and the paper would tell us how to make it, it would be useful. If it was simply a paper to show what is in the biochar from this waste stream, that would be useful also, but since it used different processes, it is not obvious to the reader that this is to show that. If this is the case, it should state something to the effect that although there has been work on biochar from several sources, wood etc., nothing has been done on mushroom compost of this particular species, therefore in this paper we will show what is in it. The authors allude to this by comparing their results to biochars made from other waste streams, but the reader does not know if one is better than another or what properties constitute good biochar and in which scenarios.

If farmers already know what they want in particular situations and the point was to show the different results that are possible using different amounts of time and temperatures, this would be useful, but if the authors know this, it should be stated and some examples given.

Another possibility is that this is a preliminary study on biochar made from a material that is a large disposal problem. This is alluded to, but what is the volume of the material? Are there environmental issues that have happened or very high disposal costs that are affecting growers? If so, examples should be given. The value of the nutrients compared to other fertilizers, especially inorganic fertilizers. In this case, it should also be presented as a preliminary study that could take researchers down other paths and suggest what these paths are. This is stated in the introduction, but it needs to be followed up on later in the text.

It would be interesting to compare the benefits of biochar to simply spreading the mushroom compost onto fields for crop production. There is much literature on using mushroom composts as fertilizers. biochar is the same, but the organic matter has mostly been destroyed. There may be a tradeoff between the two, for example unadulterated compost may be better for crop growth, but the economic cost of hauling it makes it not worth doing unless it is reduced to biochar.

Overall, this has the potential to be a good paper, it just needs to be set in the proper context, the writing needs to be clearer and grammatical errors corrected. Most of the work needs to be done on the Discussion/Conclusions sections in order to have it answer the questions posed in the Introduction. Presently, much of the Discussion/Conclusions sections reads as though it belongs in the results section.

Author Response

Thank you for your kind suggestions! 

We have honestly tried to revise the paper following your suggestions and changes have been made according to your kind comments. Your major concern regarding the use of term like waste mushroom substrate has been addressed in the response file. Moreover, I have eliminated the word industrial from the title section and paper text. 

The discussion section has been improved giving some discussion regarding use of nutrients recycling using biochar in soil and grammar errors have been checked.

We are grateful to you for your precious comments and time!

Reviewer 2 Report

My comments are in the attached pdf file.

Author Response

We are grateful to you for your constructive comments and suggestion. Your suggestions have helped us to improve our paper in a great way. I have responded to your comments in the file below!

Looking forward to your kind approval!
